# From ultraviolet-B to red photons: Effects of end-of-production supplemental light on anthocyanins, phenolics, ascorbic acid, and biomass production in red leaf lettuce

Yilin Zhu[1], Bhimanagouda S. Patil [1,2], Shuyang Zhen [1]*

**1** Department of Horticulture Sciences, Texas A&M University, College Station, Texas, United States of America, **2** Vegetable and Fruit Improvement Center, Texas A&M University, College Station, Texas, United States of America

* shuyang.zhen@tamu.edu

## Abstract

Plants possess an array of photoreceptors, such as UVR8, cryptochromes, and phytochromes, that perceive the spectral quality of light and regulate plant morphology, growth, and physiology. The use of light-emitting diodes enables the application of targeted light spectra to elicit specific plant responses during cultivation. However, there is a lack of comparative studies evaluating the effects of different spectral regions within the same crop. We comprehensively quantified how various light spectra, ranging from ultraviolet-B to red, affect plant growth and the accumulation of beneficial phytochemicals, including anthocyanins, phenolics, and ascorbic acid, in red leaf lettuce (*Lactuca sativa*) cv. Red Salad Bowl and Rouxai. Plants were grown under a background white LED light of 200 µmol m$^{-2}$ s$^{-1}$ for 16 hours per day (control), and supplemented with red (peak at 659 nm), blue (444 nm), violet (404 nm), ultraviolet-A (UVA; 368 nm) radiation at 60 µmol m$^{-2}$ s$^{-1}$, or ultraviolet-B (UVB; 309 nm) radiation at 3 µmol m$^{-2}$ s$^{-1}$ during the last 7 days of a 28-day production period (end-of-production stage, EOP). For both lettuce cultivars, red, blue and UVB treatments significantly enhanced leaf anthocyanin content compared to the control, with UVB being the most effective despite its low application dosage, followed by the blue and red light treatments. UVB radiation significantly increased total phenolic content in both cultivars (by 80%–99.1% compared to the control), while blue light treatment increased total phenolics by 31.4% in 'Red Salad Bowl' only. However, supplemental UVB radiation did not affect total ascorbic acid in either cultivar; the other EOP treatments (red to UVA) increased total ascorbic acid by 19%–35% in 'Red Salad Bowl' but had no significant effects in 'Rouxai'. Notably, crop yield under the UVB treatment was the lowest in both cultivars, with 8.9%–49% lower shoot fresh weight compared to other treatments. In contrast, the violet light treatment resulted in the highest leaf area and shoot biomass in both lettuce cultivars, although it was not effective in

**Data availability statement:** All relevant data are within the paper.

**Funding:** This research was funded by USDA-NIFA Hatch Project TEX09862, Accession No. 1026236 (SZ). This work was partially supported by USDA-NIFA-SCRI-2017-51181-26834, USDA-NIFA-2024-51181-43464 (BP) through the Vegetable and Fruit Improvement Center and Institute for Advancing Health Through Agriculture of the Texas A&M University. The sponsors or funders had no role in the study design, data collection and analysis, decision to publish, or preparation of the manuscript.

**Competing interests:** The authors have declared that no competing interests exist

enhancing anthocyanins and total phenolics. Our result indicated that there is often a tradeoff between nutritional quality and crop yield, and specific light spectra can be strategically used to enhance nutritional quality or biomass. Low-intensity UVB was the most effective at maximizing anthocyanins and total phenolics, followed by blue light, while supplemental violet light most significantly enhanced lettuce leaf expansion and biomass compared to other light spectra.

## 1. Introduction

Crop growth, yield, and nutritional quality are profoundly impacted by abiotic environmental factors, such as temperature, humidity, and light intensity. Open-field farming faces increasing challenges posed by the more frequent occurrence of extreme weather conditions. Advancements in controlled environment agriculture (CEA) technology enable precise environmental modifications to enhance crop quality and yield while improving the efficiency of resource utilization, including land, water, and fertilizer [1–4]. This presents an opportunity to complement traditional agriculture and mitigate the impact of climate change on crop production.

Nonetheless, CEA crop production requires further optimization to maximize its full potential. The high energy consumption of CEA production presents significant hurdles to its economic viability and widespread adoption. Indoor farming, in particular, relies entirely on electric lighting for crop growth and photosynthesis. Compared to field conditions, where light intensity can reach up to 2000 µmol m$^{-2}$ s$^{-1}$, the light intensities in indoor farms are typically much lower, ranging between 150 and 300 µmol m$^{-2}$ s$^{-1}$ [5]. Adequate lighting is essential for biomass production and the accumulation of beneficial phytochemicals such as phenolics and anthocyanins. Those phytochemicals increase crop resilience against environmental stressors [6–8] and offer antioxidative benefits, which are linked to human health advantages such as reduced free radicals and decreased inflammation [9–12]. High light intensity has been shown to promote the production of phenolic compounds in lettuce (*Lactuca sativa*) [13], while insufficient light can reduce yield and anthocyanin biosynthesis in red lettuce, leading to lower nutritional value and morphological appeal [14–16].

Ascorbic acid (ASC), commonly known as vitamin C, is crucial for collagen network integrity, and its deficiency can cause scurvy [17,18]. In plants, ascorbic acid scavenges reactive oxygen species to protect plants from oxidative stress [19]. Light exposure can enhance ascorbic acid production in plants by increasing GDP-1-galactose phosphorylase activity, and low light intensity may lead to lower ascorbic acid levels [20]. However, despite the benefits of higher light intensity on crop growth and nutritional quality, providing high light in CEA facilities remains challenging due to the higher capital investment requirement and high operating costs.

Currently, light-emitting diodes (LEDs) are the most energy-efficient electric lighting options available for indoor crop production [21,22]. An increasing number of studies have shown that targeted narrow-band LED lighting may be more effective in enhancing plant quality than full-spectrum white light [23,24]. However, as different

spectral regions differentially stimulate photoreceptors such as UV Resistance Locus 8 (UVR8), cryptochromes, and phytochromes, they influence not only phytochemical profiles but also plant morphology and biomass [25]. Strategic selection of the light spectrum for indoor crop lighting can thus elicit specific desirable traits to meet different production goals.

A growing body of research has focused on understanding the effects of specific light spectra on plant growth, photosynthesis, morphology, and secondary metabolism. Red light (600–700 nm) efficiently drives photosynthesis [26,27] and can affect plant growth and morphology through regulating phytochrome activities [28]. Blue light (400–500 nm) interacts with phototropin and cryptochrome photoreceptors, which modulate stomatal opening, photosynthesis, and other growth and developmental processes [29,30]. In particular, the activation of cryptochromes by blue light can inhibit stem elongation and leaf expansion, thereby reducing canopy photon capture and crop yield [31]. By contrast, active cryptochromes can also promote the production of phytochemicals, including anthocyanins [32], phenolics, flavonoids [33], and ascorbic acid [34,35]. Several studies have found that both red light and blue light can increase anthocyanin content when applied separately or in combination, although they tend to differ in effectiveness and also differently affect plant yield. Izzo et al. [36] found that, when growing lettuce cultivars Waldmann's Green and Outredgeous under a mixture of blue and red light at an intensity of 200 µmol m$^{-2}$ s$^{-1}$, plant growth decreased with increasing proportion of blue light. Shoji et al. [37] reported that anthocyanin levels in red leaf lettuce increased with increasing blue to red ratio. Owen and Lopez [38] observed that end-of-production (EOP) supplementation of monochromatic blue light, red light, or a combination of red and blue (50:50) at 100 µmol m$^{-2}$ s$^{-1}$ for 16 hours per day all led to enhanced red coloration in red leaf lettuces compared to the untreated control. However, the relative effectiveness of different spectra at enhancing anthocyanins was dependent on the application duration and the lettuce variety [38].

High energy ultraviolet-B radiation (UVB; 280–320 nm) is perceived by the photoreceptor UVR8 [39] and suppresses plant stem elongation, leaf expansion and can even damage DNA [40,41]. However, UVB is also known for its potency in promoting the production of beneficial secondary metabolites [42–45]. Lee et al. [46] showed that low intensity EOP UVB treatment applied at 1.97 W m$^{-2}$ for 2 hours per day over 4 days significantly enhanced anthocyanins in lettuce 'Two Star' compared to higher intensity ultraviolet-A (UVA; 320–400 nm) applied at 8.11 W m$^{-2}$ for 24 hours per day over 5 days.

UVA radiation shares the same flavoprotein photoreceptors with blue light [30]. UVA photons have higher energy content than blue photons, yet they tend to be less effective at activating flavoprotein photoreceptors [47]. This may lead to less inhibition of elongation and expansion growth of stems and leaves under UVA compared to blue light. Kelly and Runkle [48] found that applying 30 µmol m$^{-2}$ s$^{-1}$ of EOP supplemental UVA radiation (peak at 386 nm) for 20 hours per day enhanced dry biomass, phenolics, and anthocyanins in red leaf lettuce 'Rouxai'. Additionally, the supplemental UVA treatment was similarly effective in enhancing the production of these phytochemicals as supplemental blue light (peak at 449 nm) [48]. In contrast, Zhang et al. [33] reported that supplemental UVA (peak at 370 nm) resulted in higher leaf area in tomato (*Solanum lycopersicum*) 'Moneymaker' compared to blue light (450 nm) applied at the same intensity, but resulted in lower contents of phenolics and total flavonoids compared to blue light.

The spectral region at the interface of blue and UVA is further known as violet light. An efficient violet light LED with a peak around 404 nm has become available for horticultural production and is often marketed as a UV LED due to its photon emission in the UVA region as well as the lack of efficient LEDs that emit shorter-wavelength UV photons. The photon emission of the violet LEDs mostly falls between 380 and 425 nm, a region less effective for cryptochrome activation compared to standard blue light with a peak around 445 nm [49]. Zhang et al. [50] reported that supplemental UVA radiation (peak at 370 nm) and violet light (peak at 400 nm) applied at 20 µmol m$^{-2}$ s$^{-1}$ resulted in greater leaf area in tomato 'Moneymaker' compared to supplemental blue light treament. Zhen et al. [51] found that partially replacing blue with violet light (peak at 404 nm) resulted in higher leaf area and dry mass in cucumber 'Straight Eight'. However, a high dosage of violet light caused severe photoinhibition in lettuce 'Rex', a crop with low photoprotecive capacity. Despite these previous findings, our understanding of plant responses to violet and UVA photons remains limited.

Most previous studies on light spectral effects only examined one or two light spectral regions at a time, and there is a lack of comparative studies that evaluate the effects of a wider range of spectral regions in the same crop. Additionally, little information is currently available on the effects of UVA and violet photons. The objective of this study was to comprehensively quantify and compare the effects of different EOP light spectra — red, blue, violet, UVA, and UVB — on crop yield and nutritional quality, specifically anthocyanins, phenolics, and ascorbic acid contents in the red leaf lettuce cultivars Red Salad Bowl and Rouxai. Additionally, we aimed to investigate the potential trade-offs between yield enhancement and phytochemical production under different light spectra.

## 2. Materials and methods

### 2.1. Plant materials and growing conditions

Two red leaf lettuce cultivars, Red Salad Bowl and Rouxai, were seeded in 0.9 L containers ($10 \times 10 \times 9$ cm) filled with a soilless growing medium (Pro Mix LP15; Premier Tech Horticulture, Quakertown, PA, USA). Right after seeding, the containers were moved into two walk-in growth chambers, each sized at $2.9 \times 1.4 \times 2.4$ m ($l \times w \times h$) (Environmental Growth Chambers, Chagrin Falls, OH, USA). When the first true leaves emerged, the seedlings were assessed for uniformity and thinned to one plant per container.

During the first week, plants were fertigated with a nutrient solution containing the following nutrient concentrations (in mg $L^{-1}$): 100 N, 10.4 P, 79 K, 29.2 Ca, 7.6 Mg, 10 S, 0.13 B, 0.13 Cu, 0.53 Fe, 0.25 Mn, 0.05 Mo, and 0.25 Zn. This solution was prepared using Peters Excel 21-5-20 Multi-Purpose water-soluble fertilizer (21N-2.18P-16.6K; The Scotts Company, Marysville, OH, USA) and Epsom salt ($MgSO_4$). Subsequently, the nutrient solution concentration was increased to 150 mg $L^{-1}$ of N, with proportional increases in the concentrations of other nutrient elements. The growth chambers maintained a daytime temperature of 25°C from 0600 to 2200 h and a nighttime temperature of 22°C from 2200 to 0600 h, with a constant relative humidity of 50%.

### 2.2 Light spectral treatments

Each of the two growth chambers was divided into three separate sections using reflective barriers, resulting in a total of six individual sections. Each section was randomly assigned one of six light spectral treatments, created using LEDs: 1). A control treatment that received 200 µmol $m^{-2}$ $s^{-1}$ of white light (peak wavelengths at 451, 605, and 659 nm) for 16 h per day (0600–2200 h) throughout a 28-d production period ($W_{200}$; control), 2). End-of-production (EOP; the final 7 days of production) supplemental red light (peak at 659 nm) applied at 60 µmol $m^{-2}$ $s^{-1}$ for 16 h per day (EOP $R_{60}$), 3). EOP supplemental blue light (peak at 444 nm) applied at 60 µmol $m^{-2}$ $s^{-1}$ for 16 h per day (EOP $B_{60}$), 4). EOP supplemental violet light (peak at 404 nm) applied at 60 µmol $m^{-2}$ $s^{-1}$ for 16 h per day (EOP $V_{60}$), 5). EOP supplemental ultraviolet-A (peak at 368 nm) light applied at 60 µmol $m^{-2}$ $s^{-1}$ for 16 h per day (EOP $UVA_{60}$), and 6). EOP supplemental ultraviolet-B light (peak at 309 nm) applied at 3 µmol $m^{-2}$ $s^{-1}$ for 16 h per day (EOP $UVB_3$) (Fig 1). High-energy UVB radiation was applied at a much lower intensity than the other four light spectra (UVA, violet, blue, and red) to prevent damage to the plants, as extensive previous research has shown that UVB, though present in smaller quantities in natural sunlight, is significantly more potent compared to longer-wavelength photons. The UVB treatment at 3 µmol $m^{-2}$ $s^{-1}$ corresponded to 1.15 W $m^{-2}$, calculated based on Planck's equation ($E = hc/\lambda$; photon energy is inversely proportional to wavelength). The energy flux for the other treatments was 19.5 W $m^{-2}$ for UVA, 17.9 W $m^{-2}$ for violet, 16.1 W $m^{-2}$ for blue, and 11 W $m^{-2}$ for red light.

Plants grown in all five EOP supplemental spectral treatments received the same background white light as the control treatment. The white, red, blue, and violet LEDs were manufactured by Fluence Bioengineering (model Ray 22; Austin, TX, USA) and the UVA and UVB LEDs were custom made by Technical Consumer Products (50-W fixtures; length: 61 cm; width: 5.5 cm; Aurora, OH, USA).

Photon flux density in each treatment was measured at the canopy level (72 cm below the LEDs) at twelve locations using a spectroradiometer (PS300; Apogee Instruments, Logan, UT, USA). The spectral ratio of blue (401–500 nm), green

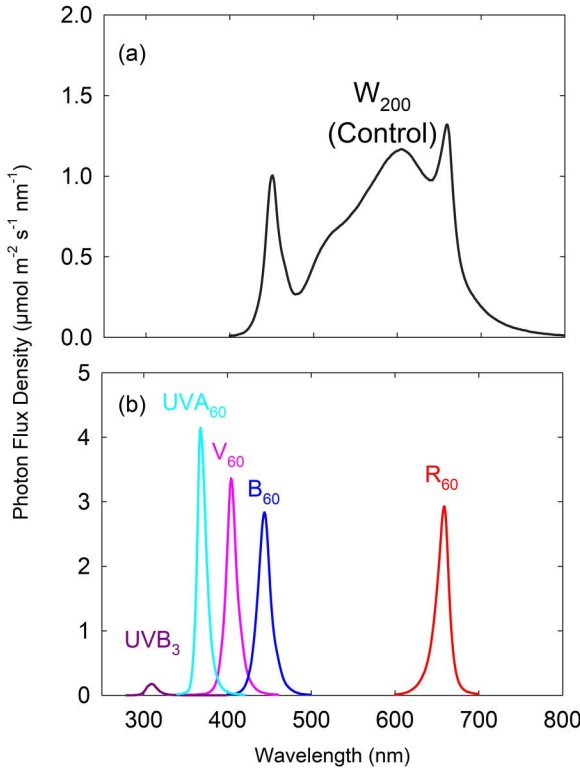

**Fig 1. Spectral distribution of the white background light (peaks at 451, 605, and 659 nm) and end-of-production (EOP) supplemental ultraviolet-B (UVB; peak at 309 nm), ultraviolet-A (UVA; peak at 368 nm), violet (V; peak at 404 nm), blue (B; peak at 444nm) and red (R; peak at 659 nm) light.** The white background light was applied at 200 µmol m⁻² s⁻¹ for 16 h/day throughout the 28-d production period ($W_{200}$; control). The EOP supplemental UVB radiation was applied at 3 µmol m⁻² s⁻¹ for 16 hours per day during the last seven days of the production period, while the EOP UVA, V, B and R lights were applied at a higher intensity of 60 µmol m⁻² s⁻¹. The subscript number after each waveband indicates its photon flux density in µmol m⁻² s⁻¹.

(501–600 nm), red (601–700 nm), and far-red (701–800 nm) in the white LEDs was 16:40:41:3; the white LEDs did not emit ultraviolet photons (Fig 1a). The daily light integral (DLI; integrated from 300 to 800 nm) in the control treatment was 11.5 mol m⁻² d⁻¹ throughout the 28-d production period. The DLI in EOP $R_{60}$, $B_{60}$, $V_{60}$ and $UVA_{60}$ supplemental light treatments increased from 11.5 to 15.0 mol m⁻² s⁻¹ during the last 7 days of production. The DLI increased slightly from 11.5 to 11.7 mol m⁻² s⁻¹ in the EOP $UVB_3$ treatment during the last 7 days of production.

Additionally, a temperature and humidity sensor (Model EE08-SS; Apogee Instruments) was placed in each chamber to measure temperature and humidity every 30 seconds, with data recorded by a data logger (CR1000X; Campbell Scientific, Logan, UT). Mixing fans were installed in each chamber to increase air circulation and ensure temperature uniformity within each chamber.

Six plants of each lettuce cultivar were placed under each light treatment (experimental plants). An additional eight plants were included in each treatment as bordering plants. The experimental plants within each treatment were rotated every two days to minimize any effect due to small spatial variations in light intensity. The average light intensity within each treatment area was very close to the target light intensity (within 0.01 µmol m⁻² s⁻¹ for UVB radiation and within 1.0 µmol m⁻² s⁻¹ for all other light spectra). The standard deviation of light intensity within the treatment area was within 7–14 µmol m⁻² s⁻¹ for the white light, 2.8–4.5 µmol m⁻² s⁻¹ for the EOP R, B, V, and UVA lights, and between 0.12–0.14 µmol m⁻² s⁻¹ for the UVB radiation. This study was replicated twice over time, and the spectral treatments were randomized in each replication.

## 2.3. Growth parameter measurements

At harvest, the shoot fresh mass of each plant was recorded. A leaf area meter (LI-3100C; LI-COR, Lincoln, NE, USA) was used to measure the total leaf area per plant. Following this, the plant shoots were placed in a drying oven at 80 °C for a week, after which the shoot dry mass was weighed.

## 2.4. Analyses of phytochemicals

### 2.4.1. Extraction-based quantifications of anthocyanins, chlorophylls, and total carotenoids.

To quantify leaf anthocyanin content, five representative leaf disks were sampled from light-exposed leaves of each plant using a cork borer (9.41 mm in diameter). The disks were soaked in 5 mL of extraction solution (5% 3M HCl + 15% $H_2O$ + 80% $C_2H_6O$) [52] and held at 6 °C for 16 h. The light absorbance of the extracted solution was measured at 530 nm ($A_{530}$) and 653 ($A_{653}$) nm using a spectrophotometer (GENESYS 180 UV-Visible; Thermo Fisher Scientific, Waltham, MA, USA). Anthocyanin index was determined using the equation described by Gould [52]:

$$Anthocyanin\ index\ =\ A530\ nm\ -\ A653\ nm\ \times\ 0.24$$

Chlorophylls and total carotenoids were assayed by taking three leaf disks from recently matured, light-exposed leaves of each plant with a cork borer (8.43 mm in diameter). The disks were immersed in 5 mL of dimethyl sulfoxide (DMSO) and incubated in a water bath at 65 °C until the leaf disks become transparent. The light absorbance of the extractions was determined at 480.0 nm ($A_{480}$), 649.1 nm ($A_{649.1}$), and 655.1 nm ($A_{655.1}$) using the same spectrophotometer described above. The pigment concentrations (µg $mL^{-1}$) were calculated following the equations described by Wellburn [53]:

$$Chlorophyll\ a\ (Chla)\ concentration\ =\ 12.47\ \times\ A655.1\ -\ 3.62\ \times\ A649.1$$

$$Chlorophyll\ b\ (Chlb)\ concentration\ =\ 25.06\ \times\ A649.1\ -\ 6.5\ \times\ A655.1$$

$$Total\ carotenoids\ concentration\ =\ (1000\ \times\ A480\ -\ 1.29\ \times\ Chla\ -\ 53.78\ \times\ Chlb)\ \div\ 220$$

Subsequently, the chlorophyll concentrations were converted to micromoles per square meter of leaf area (µmol $m^{-2}$) using the molar mass of chlorophyll a (893.51 g $mol^{-1}$), chlorophyll b (907.47 g $mol^{-1}$) and the leaf disk area per sample. The total carotenoid concentrations were converted to milligrams per square meter of leaf area (mg $m^{-2}$).

### 2.4.2. Image-based analysis of the normalized difference anthocyanin index.

Top-down RGB images of each plant were captured using a digital camera placed at 150 cm above the plant canopy. These RGB images were processed with an open-source Python script to calculate the normalized difference anthocyanin index (NDAI), an index developed by Kim and van Iersel [54] for anthocyanin content estimation. NDAI leverages the distinctive optical absorbance properties of anthocyanins, i.e., high absorption in the green region and low absorption in the red region. The Python algorithm isolates the plant imagery from its surroundings and quantifies the pixel intensity values for the red and green channels in the RGB images. NDAI is computed using the formula: $(I_{red} - I_{green}) \div (I_{red} + I_{green})$, where I denotes the pixel intensity [54] (Fig 2).

### 2.4.3. Vitamin C (ascorbic acid plus dehydroascorbic acid) content.

*Extraction.* Extraction and analysis of ascorbic acid (ASC) were performed according to Chebrolu [55] with slight modification. Two grams of the minced leaf material were placed into 50 mL test tubes, followed by the addition of 6 mL of a 3% meta-phosphoric acid (MPA) solution. Each sample was homogenized for 30 seconds using a homogenizer (Fisherbrand 850 Homogenizer; Thermo Fisher Scientific, Waltham, MA) and then ultrasonicated for 20 minutes (Cole-Parmer 8893 Ultrasonic Cleaner; Antylia

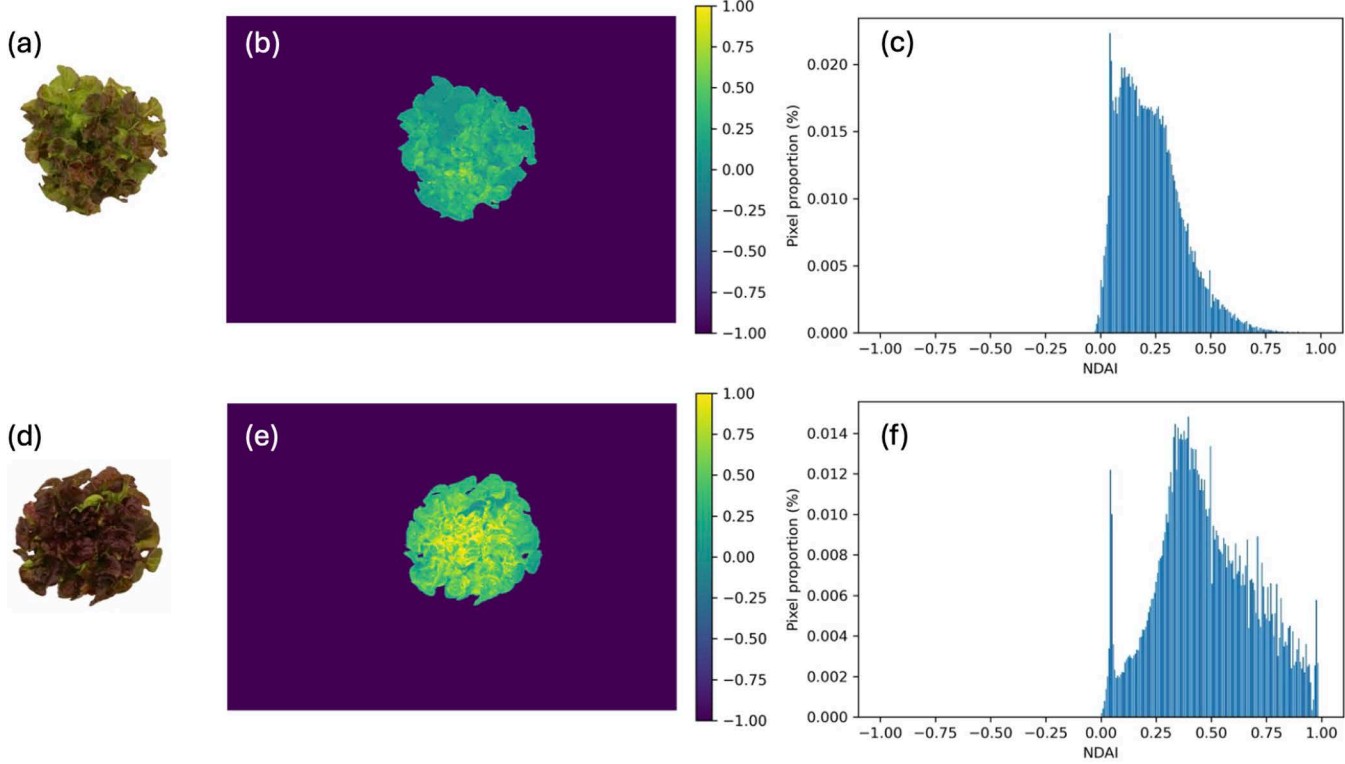

**Fig 2. Image analysis of normalized difference anthocyanin index (NDAI) following the method described by Kim and van Iersel [54].** Graphs show representative RGB images of lettuce 'Rouxai' with low (a) and high (d) anthocyanin content, and their corresponding NDAI images (b and e) and histograms (c and f).

Scientific, Vernon Hills, IL) in an ice-water bath. Subsequently, the samples were centrifuged for 12 minutes at 8228 × g at 10 °C (Eppendorf 5810R Centrifuge; Eppendorf, Hamburg, Germany), and the supernatant was decanted into new 15-mL test tubes. A 600-µL aliquot of the supernatant was then transferred into amber-colored high-performance liquid chromatography (HPLC) vials designated for ascorbic acid analysis. To determine dehydroascorbic acid (DHA) levels, 300 µL of the supernatant was mixed with an equal volume of a 0.01 M tris (2-carboxyethyl) phosphine hydrochloride (TECP) solution, and the mixture was placed into separate HPLC vials. The total vitamin C content, encompassing ASC and DHA, was quantified through HPLC analysis, with six plants per treatment sampled for this assay.

***Quantitation of ASC and DHA using UHPLC.*** ASC and DHA concentrations were quantified using an Agilent 1290 Infinity II ultrahigh performance liquid chromatography system (UHPLC; Agilent Technologies, Santa Clara, CA). Chromatographic separation was achieved on a C18 column (Eclipse Plus C18 RRHD, 1.8 µm, 2.1 x 100 mm; Agilent Technologies, Santa Clara, CA) at a flow rate of 0.1 mL min⁻¹. The binary mobile phase was composed of 0.03 M phosphoric acid (solvent A) and methanol (solvent B). The employed gradient program commenced with 0 min 100% solvent A; 3.5 min 100% solvent A; 3.6 min 25% solvent A + 75% solvent B; 4.7 min 25% solvent A + 75% solvent B; 5.0 min 100% solvent A and 5.5 min 100% solvent A. The injection volume was 4 µL and detection occurred at an absorbance wavelength of 243 nm. Calibration and quantitation were based on standard ascorbic acid solutions. DHA concentration was computed by deducting the ASC concentration from the total ASC + DHA measurement.

**2.4.4 Total phenolics.** The extraction and quantification of total phenolics were based on the protocol described by Ainsworth and Gillespie [56]. Two grams of fresh leaf material was collected from each specimen and placed into a 50 mL

test tube. To each tube, 6 mL of ethanol was added. The samples were homogenized for 30 seconds using a homogenizer (Fisherbrand 850 Homogenizer; Thermo Fisher Scientific) and subjected to ultrasonication for 20 minutes (Cole-Parmer 8893 Ultrasonic Cleaner; Antylia Scientific) within an ice-water bath. Following this, samples were centrifuged at 8228 × $g$ for 12 minutes at 10 °C (Eppendorf 5810R Centrifuge; Eppendorf, Hamburg, Germany). The supernatant was then filtered (Filter papers, 125 mm; Whatman, Maidstone, United Kingdom) and transferred into 15-mL test tubes. Each sample underwent a dual extraction process to maximize phenolic yield.

For analysis, 5 µL of the resultant extract from each sample was combined with 195 µL of nano-pure water in a culture plate (Costar 96 well cell culture plate; Corning Inc., Corning, NY). In each well of the culture plate, 20 µL of 1N phenol reagent (Folin-Ciocalteu reagent for analysis of phenols, Supelco; Supelco Inc, Bellefonte, PA) was added and allowed to react for 10 minutes at ambient temperature. Following this, 50 µL of a 280g L$^{-1}$ sodium carbonate solution was introduced to each well, and the reaction proceeded for an additional 20 minutes. The absorbance was then measured at 760 nm using a spectrophotometer (SpectraMax iD3 Multi-Mode Microplate Reader; Molecular Devices, San Jose, CA). A calibration curve for quantitation was constructed using a standard gallic acid solution with concentrations of 0, 2.5, 5, 7.5, 10, 12.5, 18.75, and 25 µg mL$^{-1}$. The resulting total phenolic content was calculated in terms of milligrams of gallic acid equivalents per gram of fresh plant mass.

## 2.5. Experimental design and statistical analysis

The experiment was replicated twice over time. The six spectral treatments were arranged in a randomized complete block design with each replicate study as a block. Data were analyzed using analysis of variance (ANOVA) in Statistical Analysis Systems software (SAS Institute, Cary, NC, USA). Mean separation was determined using Duncan's multiple-range test at a 0.05 probability level. Data analysis was carried out separately for the two lettuce cultivars.

## 3. Results

### 3.1. Leaf expansion and biomass: greatest increases under supplemental violet light but reductions under UVB radiation

The EOP spectral treatments differentially affected plant morphology, total leaf area, and shoot biomass (Figs 3 and 4). The EOP V$_{60}$ treatment resulted in the highest total leaf area (TLA) and shoot fresh weight (FW) and dry weight (DW) in both lettuce cultivars (Fig 4). Specifically, 'Red Salad Bowl' supplemented with violet light showed a 30.7% increase in TLA, a 39.9% increase in FW, and a 40.8% increase in DW compared to control. Similarly, 'Rouxai' showed a 29.5% increase in TLA, a 42.5% increase in FW and a 39% increase in DW. In contrast, EOP UVB$_3$ resulted in the lowest TLA, FW and DW across all six spectral treatments in both cultivars, with 'Rouxai' showing significant reductions (by 30.6% in FW and by 25.5% in TLA) compared to the control (Fig 4a–d, g–h).

The other EOP spectral treatments (i.e., R$_{60}$, B$_{60}$, UVA$_{60}$) either enhanced or did not cause significant changes in biomass and total leaf area compared to the control. EOP B$_{60}$ led to significant increases of 13.3% in TLA, 18% in FW and 23.9% in DW in 'Red Salad Bowl' (Fig 4a, c, and e) and a 13.2% increase in DW in 'Rouxai' (Fig 4d) compared to their respective controls. The EOP R$_{60}$ treatment did not significantly enhance TLA in either cultivar but resulted in an increase in DW by 17.8% in 'Red Salad Bowl' (Fig 4c) and 16.2% for 'Rouxai' (Fig 4d) compared to the control. The EOP UVA60 treatment did not significantly affect FW, DW, or TLA in either cultivar (Fig 4).

### 3.2. EOP UVB radiation most effectively enhanced anthocyanin production, followed by blue light and red light

All EOP supplemental light treatments led to a discernible deeper red coloration at harvest compared to the control in both lettuce cultivars (Fig 3). In 'Red Salad Bowl', the anthocyanin index, determined through pigment extraction, was significantly higher in all EOP light treatments except for EOP UVA$_{60}$ (Fig 5a). Similar trends were also observed in the image-based NDAI analysis (Fig 5c). In 'Rouxai', the anthocyanin index increased significantly under EOP R$_{60}$, B$_{60}$, and UVB$_3$

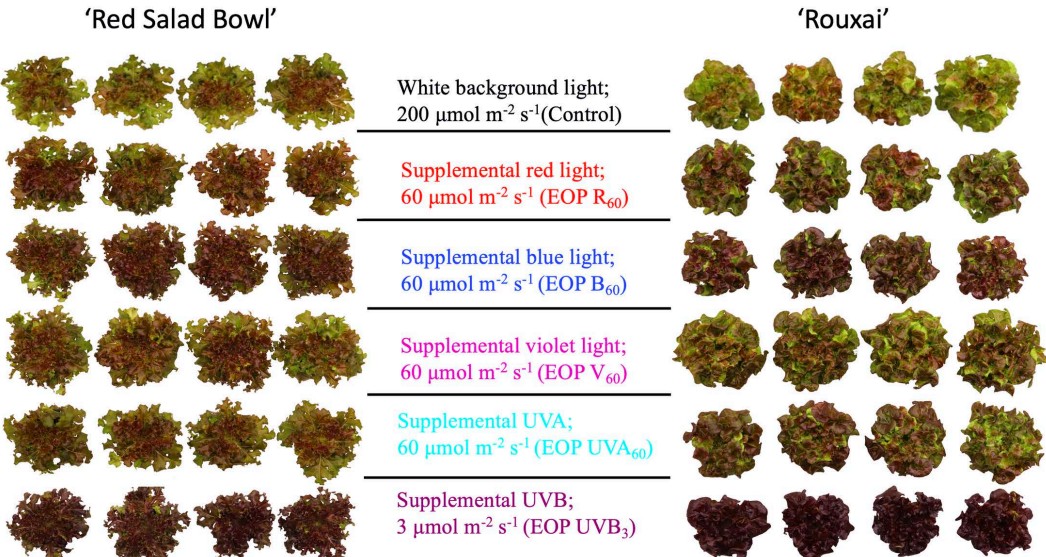

**Fig 3. Representative plants of lettuce 'Red Salad Bowl' and 'Rouxai' grown under different light spectral treatments.** EOP stands for end-of-production (the last 7 days of a 28-d production period). The photoperiod in all six treatment was 16 hours light and 8 hours dark, with EOP treatments applied throughout the 16-hour light period.

treatments compared to the control (Fig 5b), while all five EOP treatments, including $V_{60}$ and $UVA_{60}$, resulted in significantly higher values in image-based NDAI analysis (Fig 5d).

The $UVB_3$ treatment, despite its low intensity, was the most effective in enhancing anthocyanin production in both cultivars, increasing the anthocyanin index by 147% in 'Red Salad Bowl' and 314% in 'Rouxai' compared to the control (Fig 5a–b). Similarly, NDAI under the $UVB_3$ treatment increased by 141% in 'Red Salad Bowl' and 255% in 'Rouxai' compared to the control (Fig 5c–d). Following $UVB_3$, EOP $B_{60}$ was the second most effective treatment. Under $B_{60}$, anthocyanin index and NDAI increased by 69.4% and 70%, respectively, in 'Red Salad Bowl', and by 77.8% and 111%, respectively, in 'Rouxai' (Fig 5a–d).

Additionally, EOP $R_{60}$ and $V_{60}$ also enhanced anthocyanin levels in 'Red Salad Bowl', showing increases of 28.2% and 18.3% in anthocyanin index, respectively. In 'Rouxai', EOP $R_{60}$ led to a 41.2% increase in anthocyanin index, while EOP $V_{60}$ showed no significant effects (Fig 5b). The EOP $UVA_{60}$ treatment did not significantly affect the anthocyanin index in either cultivar (Fig 5a–b). The magnitudes of the spectral treatment effects were generally greater when estimated using image-based NDAI analysis. EOP $R_{60}$ and $V_{60}$ significantly increased NDAI values by 50% and 40%, respectively in 'Red Salad Bowl' (Fig 5c). In 'Rouxai', EOP $R_{60}$, $V_{60}$, and $UVA_{60}$ also increased NDAI values in 'Rouxai' by 63%, 41%, and 46%, respectively, compared to the control (Fig 5d).

A strong linear correlation was found between the extraction-based anthocyanin index and the NDAI obtained through image analysis in both cultivars. The regression equations were: NDAI = 0.3076 × Anthocyanins Index + 0.0815 in 'Red Salad Bowl' (P < 0.0001; $r^2$ = 0.84) (Fig 5e), and NDAI = 0.2515 × Anthocyanins Index + 0.1371 in 'Rouxai' (P < 0.0001; $r^2$ = 0.94) (Fig 5f).

### 3.3. Total phenolics content increased under UVB radiation

The EOP $UVB_3$ treatment significantly increased total phenolic content in both cultivars compared to the control, by 80% in 'Red Salad Bowl' and by 99% in 'Rouxai' (Fig 6). Other EOP spectral treatments, however, were not effective in enhancing phenolic content in either lettuce cultivar, with the exception of EOP $B_{60}$, which resulted in a 31.4% increase in total phenolic content in 'Red Salad Bowl' (Fig 6a).

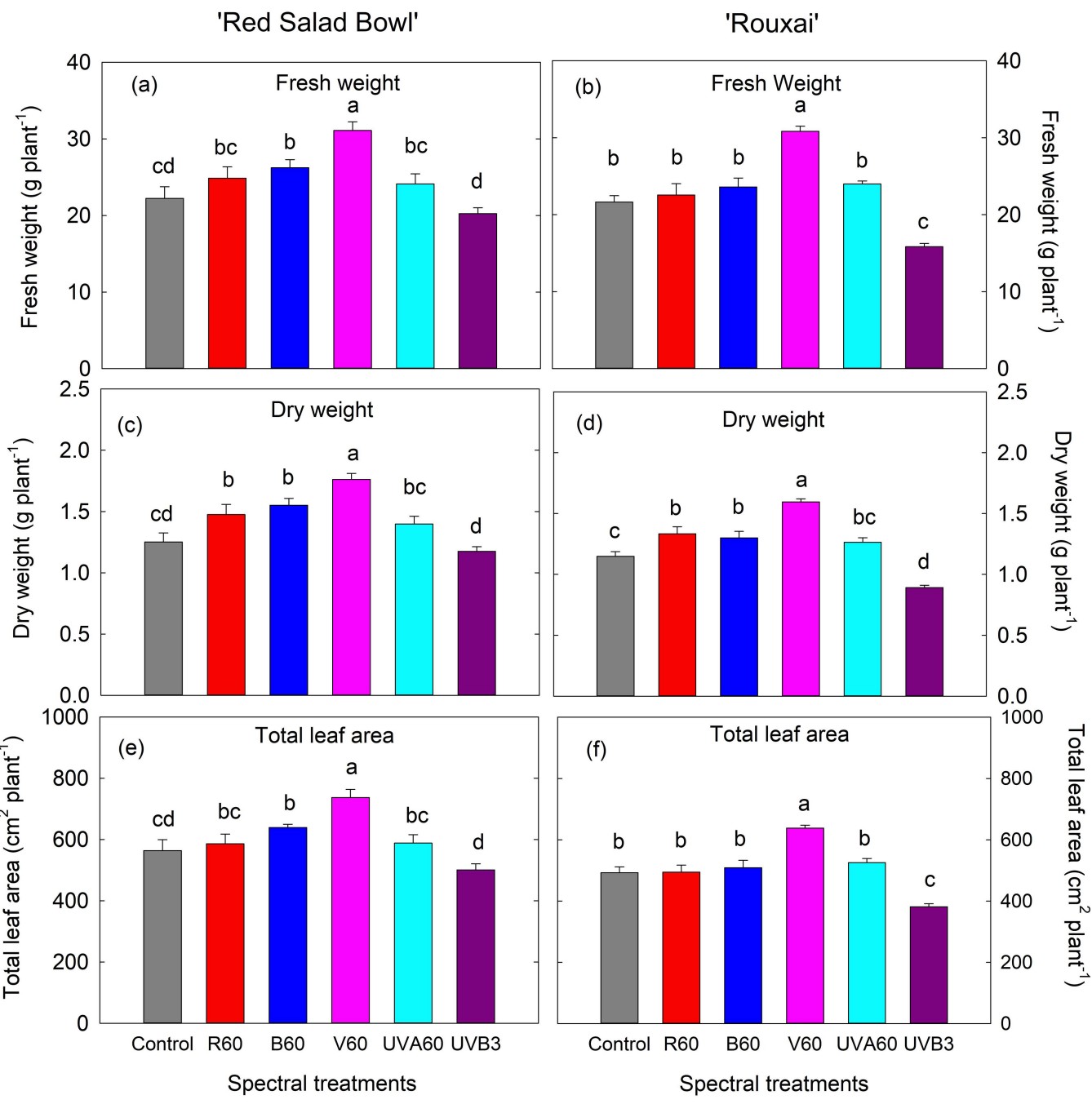

**Fig 4. Shoot fresh weight (a and b), shoot dry weight (c and d), and total leaf area (e and f) of lettuce 'Red Salad Bowl' (a, c, and e) and 'Rouxai' (b, d, and f) under different spectral treatments.** Within each lettuce cultivar, different letters indicate significance at $P < 0.05$ among the light treatments. Error bars represent SE (n = 12; 6 plants per treatment x 2 blocks). See Fig 1 legend for detailed information on the spectral treatments.

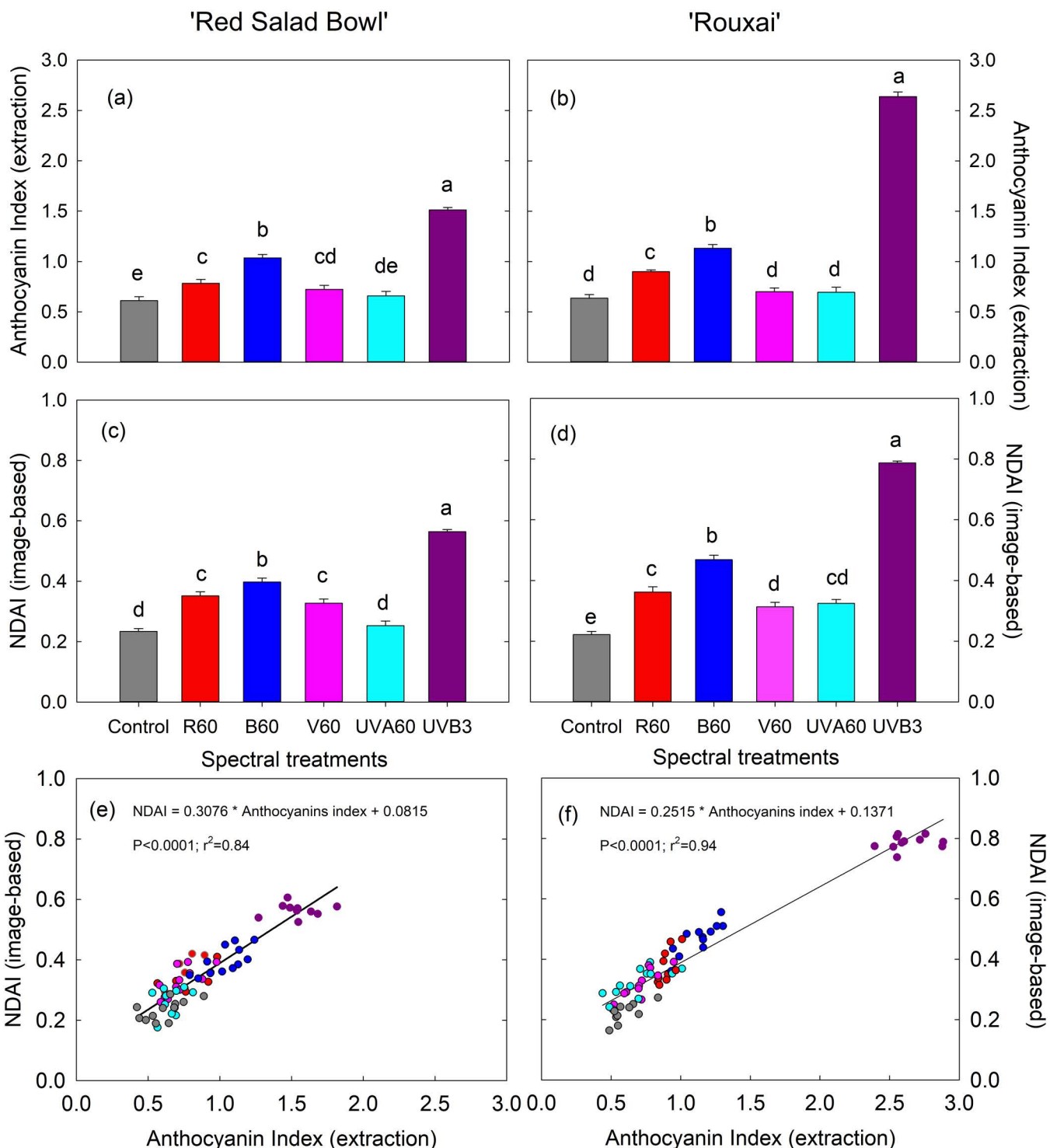

**Fig 5. Anthocyanin index (pigment extraction analysis) (a and b) and normalized difference anthocyanin index (NDAI; image-based analysis) (c and d) in lettuce 'Red Salad Bowl' (a and c) and 'Rouxai' (b and d) grown under different spectral treatments.** Within each lettuce cultivar, different letters indicate significance at $P<0.05$ among the spectral treatments. Error bars represent SE ($n=12$; 6 plants per treatment x 2 blocks). The image-based NDAI showed strong linear correlation with the extraction-based anthocyanin index in both 'Red Salad Bowl' (e) and 'Rouxai' **(f)**. See Fig 1 legend for detailed information on the spectral treatments.

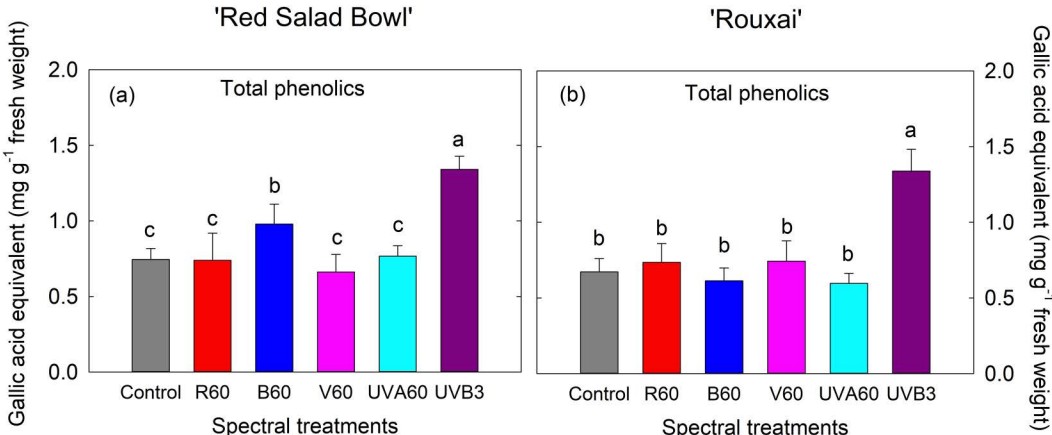

**Fig 6. Total phenolic contents of lettuce 'Red Salad Bowl' (a) and 'Rouxai' (b) in units of gallic acid equivalents (mg g$^{-1}$ fresh weight).** Within each lettuce cultivar, different letters indicate significance at $P < 0.05$ among the spectral treatments. Error bars represent SE (n = 12; 6 plants per treatment x 2 blocks). See Fig 1 legend for detailed information on the spectral treatments.

### 3.4. UVB and blue light more effectively increased total carotenoids and chlorophylls

At harvest, EOP UVB$_3$ treatment resulted in the highest total carotenoid content in both lettuce cultivars (with a 27.6% increase in 'Red Salad Bowl' and 58.4% increase in 'Rouxai' compared to the control), followed by the B$_{60}$ treatment (Fig 7a and b). EOP V$_{60}$ and UVA$_{60}$ treatments resulted in increased carotenoid content compared to the control in 'Red Salad Bowl' only, while EOP R$_{60}$ had no significant effect on carotenoid content in either cultivar (Fig 7a and b). Similar patterns were found in the contents of chlorophyll *a*, *b*, and total chlorophyll (Fig 7c and d).

### 3.5. Red, blue, violet, and UVA treatments enhanced ascorbic acid content in a cultivar-dependent manner

In 'Red Salad Bowl', both EOP R$_{60}$ and B$_{60}$ significantly enhanced ASC content by 25% compared to control (Fig 8a). Additionally, EOP R$_{60}$ and UVA$_{60}$ light enhanced DHA content by 59% and 54%, respectively, compared to the control (Fig 8a). EOP R$_{60}$, B$_{60}$, V$_{60}$ and UVA$_{60}$ significantly increased total ascorbic acid content by 35%, 24%, 19% and 30%, respectively, compared to control (Fig 8a). The EOP UVB$_3$ treatment did not significantly affect ASC, DHA, or total ascorbic acid content.

For 'Rouxai', EOP light treatments generally did not significantly affect ASC content, except for the EOP UVB$_3$ treatment, which decreased the ASC level by 18% compared to control (Fig 8b). None of the EOP treatments resulted in significant differences in DHA and total ascorbic acid levels in 'Rouxai' (Fig 8b).

## 4. Discussion

### 4.1. Light spectral effects on secondary metabolite accumulation are phytochemical-specific.

**4.1.1. EOP UVB, blue, and red light effectively promoted anthocyanin production.** Anthocyanins are responsible for the red-purple color of red leaf lettuce. With their antioxidant and anti-inflammatory effects, they can also be beneficial for human health [57,58]. Therefore, enhancing anthocyanins in red leaf lettuce through applying supplemental lighting can potentially increase its market value and offer additional health benefits to humans. Our results showed that applying EOP UVB light at a low intensity (3 μmol m$^{-2}$ s$^{-1}$) substantially enhanced anthocyanin accumulation in both 'Red Salad Bowl' and 'Rouxai'. This is consistent with previous findings that UVB radiation can promote anthocyanin production by upregulating genes involved in the anthocyanin biosynthesis pathway, such as chalcone synthase (CHS), flavanone

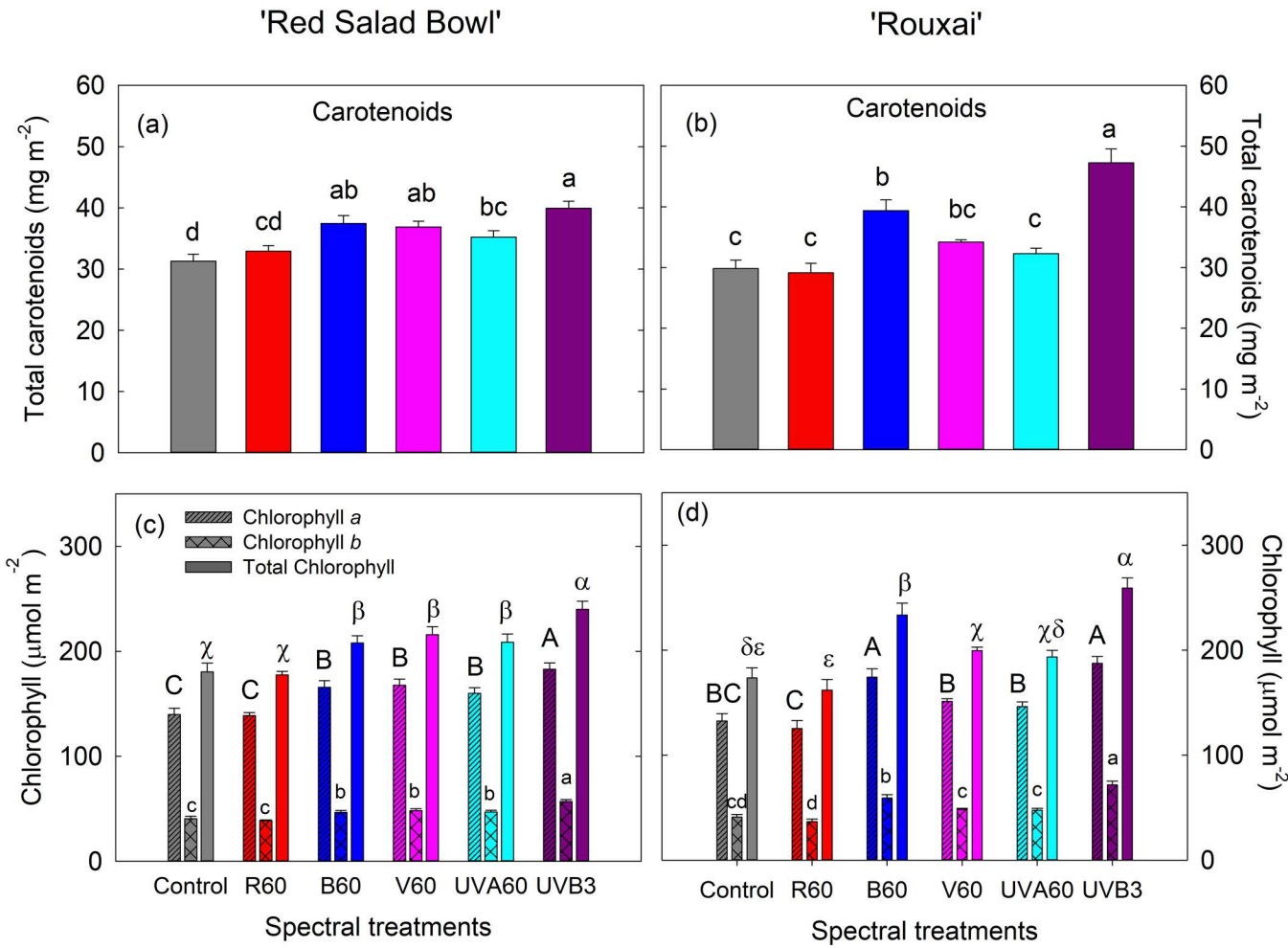

**Fig 7. Total carotenoids (a and b) and chlorophyll contents (c and d) in lettuce 'Red Salad Bowl' (a and c) and 'Rouxai' (b and d) grown under different spectral treatments.** Total chlorophyll content was calculated as the sum of chlorophyll *a* and *b* contents. Within each lettuce cultivar, different letters indicate significance at $P < 0.05$ among the light treatments. Error bars represent SE (n = 12). See Fig 1 legend for detailed information on the spectral treatments.

3-hydroxylase (F3H), and dihydroflavonol 4-reductase (DFR) [59–61]. Jenkins [62] further indicated that plants can respond to low-level UVB through UVB-specific signaling pathways that produce photoprotective phytochemicals, including anthocyanins, which may help explain the effectiveness of low-intensity UVB in inducing anthocyanin production. Rodriguez et al. [63] reported that even very low level UVB from cool white fluorescent light (0.11 W m$^{-2}$ UVB for 16 h per day) increased anthocyanin accumulation in lettuce 'Carmoli'. Note that the UVB photon flux density of 3 µmol m$^{-2}$ s$^{-1}$ used in our study is equivalent to an energy flux density of 1.15 W m$^{-2}$. Furthermore, the high-energy UVB radiation was significantly more potent at enhancing the accumulation of anthocyanins, total phenolics, chlorophylls and carotenoids than the lower-energy UVA, V, B, and R light. UVB was applied at 20 times lower intensity but resulted in the highest contents of these phytochemicals (Figs 5–7). Given the high efficacy of UVB applied at 3 µmol m$^{-2}$ s$^{-1}$, there is potential to apply UVB at even lower intensities to achieve adequate levels of anthocyanins and other phytochemicals while reducing energy costs and the capital investment in acquiring UVB fixtures. A lower UVB dosage could also minimize the potential risk of damaging the crops. It is worth noting that 'Rouxai' accumulated substantially more anthocyanins than 'Red Salad

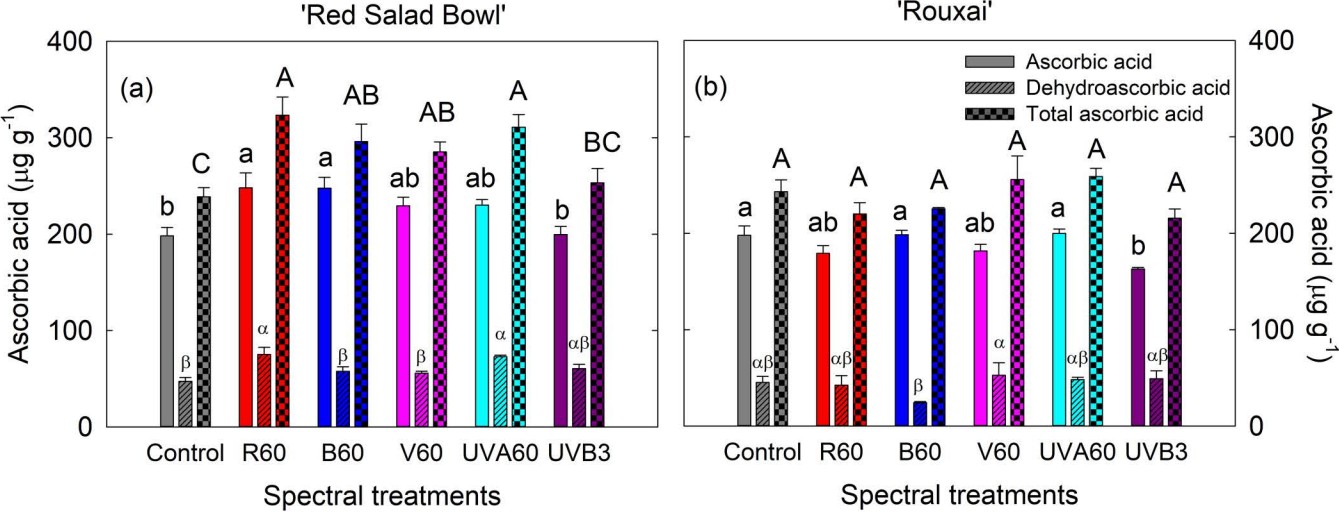

**Fig 8. Ascorbic acid, dehydroascorbic acid, and total ascorbic acid levels of lettuce 'Red Salad Bowl' (a) and 'Rouxai' (b) grown under different light spectral treatments.** Within each lettuce cultivar, different letters indicate significance at $P < 0.05$ among the spectral treatments. Error bars represent SE (n = 6). See Fig 1 legend for detailed information on the spectral treatments.

Bowl' in response to the UVB treatment, indicating that the effects of light treatments can differ among cultivars within the same species. It is possible that the lettuce cultivars differ in expression levels of key genes regulating anthocyanin accumulation [64]. Therefore, cultivar/genotype-specific lighting optimization may be necessary.

Following the UVB$_3$ treatment, EOP B$_{60}$ was the second most efficacious for enhancing anthocyanin levels. Mediated by photoreceptor cryptochromes, blue light increases the transcription of the genes encoding dihydroflavonol 4-reductase and anthocyanidin synthase, which are involved in the biosynthesis of anthocyanins [65]. Blue light has been shown to promote anthocyanin production in various crops [32,48,66]. However, comprehensive studies comparing the effectiveness of different light spectral treatments at regulating phytochemical production remain limited. Our results indicate that, when applied at the same photon flux density, supplemental blue light was more effective at enhancing anthocyanins in red leaf lettuce than longer-wavelength red light and shorter-wavelength violet and UVA radiation (Fig 5).

Red light has been shown to increase the expression of anthocyanin biosynthesis genes, including CHS, and promote anthocyanin production in crops such as strawberries and cranberries [67,68]. Our experiment showed that red light (EOP R$_{60}$) significantly promoted anthocyanins in both cultivars compared to the control, however, it was less effective than blue light.

EOP UVA$_{60}$ was the only treatment that did not significantly increase extraction-based anthocyanin index in either cultivar, although image-based NADI analysis indicated increased anthocyanin accumulation in 'Rouxai' under the UVA$_{60}$ treatment. One possible explanation is that UVA is less effective at activating flavoprotein photoreceptors [47], leading to a lower production of anthocyanins in red leaf lettuce. Our result suggests that supplemental UVA applied at an intensity of 60 µmol m$^{-2}$ s$^{-1}$ for 16 h per day may be insufficient to enhance anthocyanins in red leaf lettuce. In contrast, Kelly and Runkle [48] showed 30 µmol m$^{-2}$ s$^{-1}$ of EOP UVA (peak at 386 nm) for 20 hours per day enhanced anthocyanins in 'Rouxai'. However, it should be noted that the background light in their study comprised of a higher ratio of red light (56% red LED light plus 44% warm white LED light), in contrast to the 41% total red light in the background white light used in our study. This spectral difference could have contributed to the different responses observed. Furthermore, it is possible that the leaf disk sampling for the pigment extraction may not be fully representative despite our efforts. Nonetheless, a strong linear correlation was observed between the extraction-based anthocyanin index and the imagine-based NDAI,

which indicated that imagine-based analysis could be a reliable and more efficient method for analyzing anthocyanins in red leaf lettuce. Further research may explore higher UVA intensities and/or longer application durations to determine its effectiveness on anthocyanin production.

Note that NDAI analysis of top-down plant images provides an estimate of projected leaf area only. The canopy structure of lettuce is relatively simple; however, the NDAI values are less representative of whole-canopy anthocyanin levels in plants with many curved, overlapping, or shaded leaves. Because image-based NDAI relies on pixel intensity analysis, lighting conditions and camera settings can influence the pixel intensity values. In this study, all images were taken under consistent plant-camera orientation, lighting conditions, and camera setting. It is a useful technique for estimating canopy-level anthocyanin content and comparing treatment differences. However, NDAI values from different experimental setups may not be directly comparable. Additionally, detecting low anthocyanin levels in green leaves is difficult due to limited pigmentation contrast.

**4.1.2. EOP supplemental UVB radiation most effectively increased total phenolics.** UVB radiation has been shown to effectively promote phenolic production [69–71]. When exposed to UVB radiation, the UVB receptor UVR8 monomerizes and interacts with COP1 [39], which further mediates photomorphogenic responses such as inhibition of leaf expansion and the upregulation of the gene expression for key enzymes in the phenolic acid synthesis pathway [62,72,73]. Higher intensity UVB can cause increased reactive oxygen species (ROS) generation in plants [74,75], which, in turn, can induce increased antioxidant production [62]. Similar to the responses of anthocyanins, EOP UVB3 treatment resulted in the highest total phenolic content in both lettuce cultivars (Fig 6).

Previous studies found that blue light can also enhance phenolics [76,77]. Phenylalanine ammonia-lyase (PAL) is the first enzyme in the phenylpropanoid pathway and plays a crucial role in phenolic production [78]. Blue light was found to upregulate *PAL* gene expression and enhance PAL enzyme activity [79,80]. Consistent with these findings, we observed a significant increase in total phenolic content in 'Red Salad Bowl' under blue light (Fig 6a).

**4.1.3. Supplemental UVB radiation and blue light promoted carotenoids and chlorophylls.** EOP $UVB_3$ and $B_{60}$ treatments significantly increased the levels of carotenoids and chlorophylls in both lettuce cultivars, with UVB being more effective in enhancing chlorophyll content in 'Red Salad Bowl' and carotenoid content in 'Rouxai'. Blue light has been known to enhance the biosynthesis of chlorophylls [81] and to stimulate the transcriptional activity of genes involved in carotenoid biosynthesis [82]. As for UVB, Badmus et al. [83] found that UV radiation (mainly UVB) can boost the synthesis of several carotenoids in plants. However, the UVR8 pathway was not directly involved in carotenoid synthesis. This suggests that plants might perceive UV radiation as an indicator of environmental stress and produce more carotenoids as a defense mechanism. The impact of UVB on chlorophyll content, however, is inconsistent across studies. Smith et al. [84] observed inconsistent relationship between chlorophyll content and UVB sensitivity among lettuce cultivars; plants exposed to UVB either experienced an increase in chlorophyll content or were unaffected. Our findings support the notion that EOP UVB can enhance chlorophyll and overall carotenoid content in both cultivars of red leaf lettuce. We also observed increases in carotenoids and chlorophylls content in 'Red Salad Bowl' under EOP $V_{60}$ and $UVA_{60}$ treatments. In contrast, Caldwell and Britz [85] observed that supplemental UVA did not enhance chlorophyll and carotenoid content in red leaf lettuce, and combined UVB+UVA treatments actually decreased the contents of these pigments. This may be attributed to increased accumulation of UV-induced phenolic compounds in epidermal cells, which can shield the plant tissues from UV light. The EOP $R_{60}$ treatment did not have significant enhancement on chlorophylls and carotenoids content in either red leaf lettuce cultivar, which is similar to the findings of Lee et al. [86] and Li and Kubota [32].

**4.1.4. Supplemental UVB did not affect ascorbic acid content, while other EOP light spectra enhanced ascorbic acid content or had no affects depending on the cultivar.** Ascorbic acid serves a protective role as a reactive oxygen species (ROS) scavenger against stressors such as excessive light [87,88]. Previous research has indicated that ascorbic acid levels can be increased by exposing plants to higher light intensities [89,90]. Consequently, the enhanced total ascorbic acid levels observed in 'Red Salad Bowl' under EOP $R_{60}$, $B_{60}$, $V_{60}$, and $UVA_{60}$ treatments may be attributed to the high photosynthetic efficiency of red light. Kang et al. [91] reported that irradiating Chinese cabbage 'Dongbu' with

blue light (125 µmol m$^{-2}$ s$^{-1}$ for 16 h per day for 5 days) enhanced expression of ascorbic acid biosynthesis genes such as *BrPGI1*, *BrPMI1*, *BrPMM1*, *BrGMP1*, *BrGME1*, *BrGGP1*, and *BrGPP*. Also, blue light has been shown to boost ascorbic acid synthesis by inhibiting the PAS/LOV photoreceptor, a negative regulator of ascorbate biosynthesis [20]. Liu et al. [92] found that UVB radiation applied at 3.33 µmol m$^{-2}$ s$^{-1}$ increased ascorbic level and the expression level of genes involved in ascorbic acid metabolism such as MIOX1 and GLDH in cucumber 'Chinese long 9930'. In our experiment, the EOP UVB$_3$ treatment resulted in a decrease in ascorbic acid levels in 'Rouxai'. Given that UVB radiation induces high oxidative stress, it is reasonable to expect an increase in ascorbic acid content. It is noteworthy that 'Rouxai' showed considerably higher levels of anthocyanins and total phenolics under UVB treatment compared to other treatments (Fig 5b and d; Fig 6b). It is possible that the increased accumulation of anthocyanins and total phenolics resulted in greater light competition with chlorophylls, thereby reducing plant growth and ascorbic acid synthesis. This may also explain why the EOP R$_{60}$, B$_{60}$, V$_{60}$, and UVA$_{60}$ treatments were more effective in enhancing ascorbic acid levels in 'Red Salad Bowl' compared to 'Rouxai', as 'Red Salad Bowl' produced lower levels of anthocyanins in response to the spectral treatments.

## 4.2. Tradeoff between crop yield and nutritional quality

### 4.2.1. Supplemental violet light most significantly increased crop yield while UVB caused yield reductions.
Violet light, which is at the interface of UVA and blue light, is on average less effective than blue light in activating cryptochromes [49]. Active cryptochromes can suppress stem elongation [93–95]. Zhen et al. [51] suggested that in species and cultivars with adequate photoprotective pigments, violet light could be less inhibitive to leaf expansion than blue light, resulting in higher canopy radiation capture and biomass production when used in place of blue light. Similarly, we found that EOP V$_{60}$ treatment resulting in the highest total leaf area and shoot fresh and dry biomass in both red lettuce cultivars. Zhang et al. [50] found that replacing about 10% of the background red light with UVA (370 nm) or violet (400 nm) light resulted in the highest leaf area in tomato plants. This was followed by replacing ~10% of the red light with blue light, and finally, by applying 100% red light at 190–193 µmol m$^{-2}$ s$^{-1}$). Our results differed slightly in that supplemental violet light was more effective than UVA, while UVA and blue were similarly effective in enhancing leaf expansion and lettuce yield (Fig 4). However, we used higher intensities of supplemental UVA, violet, and blue lights with a different background light (white versus red). Also, different plant species may respond differently. Zhang et al. [33] further shown that UVA did not significantly affect photosynthetic parameters in tomato compared to blue light. Therefore, the higher biomass observed under violet light in our study was most likely due to more efficient light interception from increased leaf area. It is worth noting that, in our experiment, supplemental red light treatment (EOP R$_{60}$) led to lower fresh and dry biomass compared to violet light treatment (EOP V$_{60}$), even though red light may be used more efficiently for photosynthesis [26]. Leaf chlorophyll content was significantly lower in the EOP R$_{60}$ treatment compared to the V$_{60}$ treatment in both lettuce cultivars (Fig 7). This may have led to reduced leaf light absorption and consequently contributed to the lower biomass under the R$_{60}$ treatment compared to the V$_{60}$ treatment.

The EOP UVB$_3$ treatment was the only supplemental light treatment that caused significant reductions in both total leaf area and biomass in 'Rouxai' compared to the control. Similar reduction trends were observed under UVB$_3$ in 'Red Salad Bowl', although the responses were not statistically significant. The inhibition of leaf expansion by UVB radiation is well-documented [72,96], which can lead to reduced photon capture and biomass. Additionally, the high anthocyanin production induced under EOP UVB$_3$, particularly in 'Rouxai', could further reduce the light absorption by photosynthetic pigment chlorophylls. This trade-off between improved crop nutritional quality and potential yield reduction must be considered in commercial applications.

## 4.3. EOP supplemental light duration

Plants can rapidly accumulate phytochemicals such as phenolics in response to environmental stimuli, often within days. Therefore, EOP supplemental lighting can be an effective strategy for enhancing crop nutritional quality. However,

to achieve a significant increase in phytochemical content, an adequate light exposure duration is required. Owen and Lopez [38] recommended applying red and blue supplemental light for more than five days at a photon flux density of 100 µmol m$^{-2}$ s$^{-1}$ for 16 hours per day to enhance anthocyanin production in red leaf lettuce. Longer exposure durations tend to increase the effectiveness of EOP light treatments. For instance, Kang et al. [97] reported that mild UV-A light (35–97 µmol m$^{-2}$ s$^{-1}$) increased total phenolics content and antioxidant capacity in Sweet Basil (*Ocimum basilicum*), with greater stimulative effects observed as the exposure duration increased over a 14 day-period. Similarly, Chen et al. [98] found that 10 days of UVA exposure at 10 µmol m$^{-2}$ s$^{-1}$ for 16 h per day significantly increased total phenolic content in lettuce, whereas 5 days of exposure had no significant effect.

In this study, we used a 7-day EOP supplemental lighting period to ensure sufficient phytochemical accumulation in red leaf lettuce across the supplemental light treatments.

### 4.4. Photon efficacy of different light spectra

Although supplemental red, blue, violet, and UVA lights were applied at the same intensity (and UVB radiation was applied at a lower intensity), the electricity usage likely varied among these treatments. According to Kusuma et al. [22], the red LED package (660 nm) has the highest average photon efficacy (µmol J$^{-1}$), followed by blue (450 nm), violet (405 nm), UVA (385 nm), and UVB (310 nm) LEDs. The photon efficacy of common UVB LEDs is more than a hundred times lower than red LEDs and about ninety times lower than blue (450 nm) LEDs. Even though UVB$_3$ most effectively promoted anthocyanins and total phenolics, the higher electricity consumption, higher fixture costs, limited commercial availability, and potential yield reductions make UVB radiation a less ideal option for crop production. On the other hand, EOP B60 was the second most effective treatment for both yield improvement and the accumulation of beneficial phytochemicals such as anthocyanins and phenolics. Blue LEDs are energy efficient and widely available. Additionally, blue light has been shown to be more effective in enhancing anthocyanin accumulation than broad spectrum white LED light [23]. Therefore, monochromatic blue light emerges as a good option for end-of-production supplemental lighting for red leaf lettuce.

Since energy use depends on fixture efficacy—which varies among manufacturers—as well as operating conditions such as installation height, the number of fixtures, and the effect of energy droop, we report the relative efficiency of the light fixtures used in our study rather than absolute wattage values. Red LEDs were the most energy-efficient. Blue LEDs required 46% more energy than red LEDs to reach the same photon flux density, while violet LEDs required 4.1 times more energy. UVA LEDs, made by a different manufacturer, required 3.85 times more energy than red LEDs to deliver the same light intensity. Notably, UVB LEDs required more than 15 times the energy to deliver just one-twentieth of the photon flux density used in the other treatments.

## 5. Conclusion

Compared to lower-energy UVA, violet, blue, and red lights, EOP UVB radiation applied at much lower intensity (3 µmol m$^{-2}$ s$^{-1}$ for 16 h per day) most effectively enhanced anthocyanins, total phenolics, carotenoids, and chlorophyll in red leaf lettuce. However, high-energy UVB radiation may cause yield reductions and potentially lower vitamin C content depending on the lettuce cultivar. EOP blue light (60 µmol m$^{-2}$ s$^{-1}$ for 16 h per day) was the second most effective treatment for enhancing both crop yield and the accumulation of anthocyanins and phenolics. On the other hand, EOP violet light (60 µmol m$^{-2}$ s$^{-1}$ for 16 h per day), although less effective at improving crop nutritional quality, resulted in the greatest leaf expansion and yield. A broader range of lettuce cultivars should be evaluated under these EOP light treatments in future studies, as genotype × light environment interactions may influence the accumulation of anthocyanins and other beneficial phytochemicals.

In commercial indoor agriculture, specific light spectrum can be selected to achieve different production goals, for instance, yield increase or improved crop quality. However, it is essential to balance the need to enhance crop quality without significantly compromising yield. The fixture costs and the LED efficacy (i.e., energy consumption) of different light spectra should be considered in commercial use.

## Acknowledgments

We thank Seonghwan Kang, Samiksha Bhattarai, Dr. Yuyao Kong and Dr. Deepak Kumar JHA for their assistance with data collection.

## Author contributions

**Conceptualization:** Shuyang Zhen.

**Data curation:** Yilin Zhu.

**Formal analysis:** Yilin Zhu.

**Funding acquisition:** Bhimanagouda S. Patil, Shuyang Zhen.

**Investigation:** Yilin Zhu.

**Methodology:** Yilin Zhu, Bhimanagouda S. Patil, Shuyang Zhen.

**Project administration:** Shuyang Zhen.

**Resources:** Bhimanagouda S. Patil, Shuyang Zhen.

**Supervision:** Bhimanagouda S. Patil, Shuyang Zhen.

**Validation:** Yilin Zhu, Bhimanagouda S. Patil, Shuyang Zhen.

**Visualization:** Yilin Zhu.

**Writing – original draft:** Yilin Zhu.

**Writing – review & editing:** Yilin Zhu, Bhimanagouda S. Patil, Shuyang Zhen.

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
