## [Decision Letter · Decision Letter 0]

PONE-D-24-54525From ultraviolet-B to red photons: Effects of end-of-production supplemental light on anthocyanins, phenolics, ascorbic acid, and biomass production in red leaf lettucePLOS ONE

Dear Dr. Zhen,

Thank you for submitting your manuscript to PLOS ONE. After careful consideration, we feel that it has merit but does not fully meet PLOS ONE’s publication criteria as it currently stands. Therefore, we invite you to submit a revised version of the manuscript that addresses the points raised during the review process.

We look forward to receiving your revised manuscript.

Kind regards,

Jianhong Zhou

Staff Editor

PLOS ONE

Journal Requirements:

3. In the online submission form, you indicated that all relevant data are within the manuscript. Additional data (including raw data) are available from the authors upon request.

Reviewers' comments:

Reviewer's Responses to Questions

**Comments to the Author**

1. Is the manuscript technically sound, and do the data support the conclusions?

Reviewer #1: Yes

Reviewer #2: Yes

2. Has the statistical analysis been performed appropriately and rigorously? 

Reviewer #1: Yes

Reviewer #2: Yes

3. Have the authors made all data underlying the findings in their manuscript fully available?

Reviewer #1: Yes

Reviewer #2: Yes

4. Is the manuscript presented in an intelligible fashion and written in standard English?

Reviewer #1: Yes

Reviewer #2: Yes

5. Review Comments to the Author

Reviewer #1: This manuscript investigates how different narrow-band LEDs (UVB, UVA, violet, blue, and red applied at the end of the production cycle influence biomass production and phytochemical accumulation in two red-leaf lettuce cultivars.

Strengths

The use of two cultivars highlighted genotype-specific responses.

The study is well controlled. The data are also clearly presented and statistically analyzed with replication and an appropriate experimental design.

The research has high relevance to both the scientific community and indoor farms, as well as LED optimization in CEA.

I’m happy to see the data correlating NDAI and extracted anthocyanins.

Minor suggestions

I would like to see some discussion about the lack of energy equivalency between treatments. I understand that equipment limitations may be the reason that different LEDs were applied at different PFDs, but I would like to see some more detail in the discussion. I also think providing the total energy (W m⁻²) for each light treatment (especially the UVB treatment) would be useful.

Please explain why 7 days was picked for the EOP treatment duration. Previous research on EOP lighting has used shorter durations and seen positive benefits.

I’d like to see a bit more discussion on why there might be some cultivar differences.

What leaves/where on the leaf was the tissue for chemical analysis extracted from? Was it from light-exposed tissue?

What limitations are there for NDAI? I imagine leaf angle, overlapping leaves, and other pigments could influence the results.

Specific suggestions

L82. I believe “hurtles” should be “hurdles.”

L650. “Comprising” should be “compromising.”

Reviewer #2: The manuscript PONE-D-24-54525 “From ultraviolet-B to red photons: Effects of end-of-production supplemental light on anthocyanins, phenolics, ascorbic acid, and biomass production in red leaf lettuce” by Zhu et al is a well-written manuscript and is ready for publication in my opinion. Authors clearly show that using specific light settings enhance beneficial compounds in lettuce with no yield increase. I think this research is valuable for controlled environment agriculture producers as could add benefits to producing crops like lettuce in such settings.

Although yield was not increased with the use of different lights, this characteristic in lettuce is not necessarily desirable as yield depends on markets. Sometimes, the desired product does not have to be the one with highest yield. For instance, the use of these settings could well enhance the nutritional content of baby leaf lettuce production; the market uses a mix of green and red baby leaf lettuce.

I am also wondering if in conclusions authors can a sentence or two somewhat recommending that other red leaf lettuce cultivars should be screened with their recipes. It seems that there is a genotype x environment (light setting) interaction that could be considered by plant breeders while enhancing these compounds on red leaf lettuce.

I would also like to request from authors to delete quotation marks from cultivar names when you precede the name with the word “cultivar”, “genotype”, etc.

Congratulations on a well written manuscript.

6. PLOS authors have the option to publish the peer review history of their article (what does this mean? ). If published, this will include your full peer review and any attached files.

**Do you want your identity to be public for this peer review?** For information about this choice, including consent withdrawal, please see our Privacy Policy .

Reviewer #1: No

Reviewer #2: No

---

## [Author Response · Author response to Decision Letter 1]

9 Jun 2025

Dear Editor,

We have thoroughly revised the manuscript in response to the reviewers’ comments. A point-by-point response to each comment is provided below.

In addition, we have formatted the manuscript to comply with PLOS ONE’s style requirements and reviewed the ‘funding information’ section. All relevant data are included within the manuscript, and we have reviewed and confirmed that our reference list is complete and correct.

Please let us know if additional revisions are needed. Thank you, and we sincerely appreciate your time and thoughtful consideration.

Best regards,

Yilin Zhu and Shuyang Zhen

Co-author: Bhimanagouda Patil

Review Comments to the Author

Reviewer #1: This manuscript investigates how different narrow-band LEDs (UVB, UVA, violet, blue, and red applied at the end of the production cycle influence biomass production and phytochemical accumulation in two red-leaf lettuce cultivars.

Strengths

The use of two cultivars highlighted genotype-specific responses.

The study is well controlled. The data are also clearly presented and statistically analyzed with replication and an appropriate experimental design.

The research has high relevance to both the scientific community and indoor farms, as well as LED optimization in CEA.

I’m happy to see the data correlating NDAI and extracted anthocyanins.

Response: Thank you for your comments. We appreciate your time and feedback.

Minor suggestions

I would like to see some discussion about the lack of energy equivalency between treatments. I understand that equipment limitations may be the reason that different LEDs were applied at different PFDs, but I would like to see some more detail in the discussion. I also think providing the total energy (W m⁻²) for each light treatment (especially the UVB treatment) would be useful.

Response: Thank you for the suggestion. All supplemental treatments were applied at the same PFD of 60 µmol m-2 s-1, except for the UVB treatment, which was applied at a lower PFD of 3 µmol m-2 s-1 to avoid plant damage due to the high energy content of UVB radiation. We have included the total energy flux in W m-2 for each supplemental light treatment to facilitate comparison, as UVB dosages in previous publications are often expressed in W m-2. The UVB treatment at 3 µmol m-2 s-1 corresponds to 1.15 W m-2 (calculated using the Planck’s equation). The energy flux for the other treatments was 19.5 W m-2 for UVA, 17.9 W m-2 for violet, 16.1 W m-2 for blue, and 11 W m-2 for red light. This information has been included in the manuscript.

Additionally, we have also included a discussion of energy consumption among the treatments. Energy consumption depends on fixture efficacy (which varies among manufacturers) and operating conditions such as installation height, the number of fixtures, and the effect of energy droop. Therefore, we report the relative efficiency of different light spectra used in our study rather than the absolute wattage values. In our study, red LEDs were the most energy efficient. Blue LEDs required 46% more energy than red LEDs to reach the same PFD, while violet LEDs required 4.1 times more energy. UVA LEDs, manufactured by a different company, required 3.85 times more energy than red LEDs to reach the same light intensity. Notably, UVB LEDs required more than 15 times the energy to deliver just one-twentieth of the PFD used in the other treatments. We hope this provides a useful reference for comparing the energy consumption across the treatments.

Please explain why 7 days was picked for the EOP treatment duration. Previous research on EOP lighting has used shorter durations and seen positive benefits.

Response: Thank you for the comment. We agree that EOP lighting treatments have been shown to elicit positive benefits with shorter durations. However, both previous studies and our own unpublished results suggest that longer exposure duration (e.g., 6-7 days) tend to have greater effects compared to shorter durations (e.g., 3 days). We have added a paragraph in the discussion section to explain our rationale for selecting a 7-day treatment duration.

I’d like to see a bit more discussion on why there might be some cultivar differences.

Response: Thank you for the comment. The two cultivars exhibited slight differences in their responses to the EOP treatments. For example, Rouxai had higher anthocyanin accumulation and tended to be more responsive the UVB treatments. However, the overall response patterns to the different treatments were similar between the two cultivars. It is possible that the lettuce cultivars differ in expression levels of key genes regulating anthocyanin accumulation. We have added further discussion and references regarding cultivar-specific differences in anthocyanin production.

What leaves/where on the leaf was the tissue for chemical analysis extracted from? Was it from light-exposed tissue?

Response: Thank you for the comment. We sampled leaf tissues that were directly exposed to light. We have clarified this in the experimental design section.

What limitations are there for NDAI? I imagine leaf angle, overlapping leaves, and other pigments could influence the results.

Response: Thank you for the comment. NDAI analysis of top-down plant images provides an estimate of projected leaf area only. The canopy structure of lettuce is relatively simple. The values are less representative of whole-canopy anthocyanin levels in plants with many curved, overlapping, or shaded leaves. Because image-based NDAI relies on pixel intensity analysis, lighting conditions and camera settings can influence the pixel intensity values. In this study, all images were taken under consistent plant-camera orientation, lighting conditions, and camera setting. It is a useful technique for estimating canopy-level anthocyanin content and comparing treatment differences. However, NDAI values from different experimental setups may not be directly comparable. Additionally, detecting low anthocyanin levels in green leaves is difficult due to limited pigmentation contrast. We have added a discussion of these limitations.

Specific suggestions

L82. I believe “hurtles” should be “hurdles.”

L650. “Comprising” should be “compromising.”

Response: Thank you, we have corrected the spelling errors.

Reviewer #2: The manuscript PONE-D-24-54525 “From ultraviolet-B to red photons: Effects of end-of-production supplemental light on anthocyanins, phenolics, ascorbic acid, and biomass production in red leaf lettuce” by Zhu et al is a well-written manuscript and is ready for publication in my opinion. Authors clearly show that using specific light settings enhance beneficial compounds in lettuce with no yield increase. I think this research is valuable for controlled environment agriculture producers as could add benefits to producing crops like lettuce in such settings.

Response: Thank you for your comment. We appreciate your time and feedback.

Although yield was not increased with the use of different lights, this characteristic in lettuce is not necessarily desirable as yield depends on markets. Sometimes, the desired product does not have to be the one with highest yield. For instance, the use of these settings could well enhance the nutritional content of baby leaf lettuce production; the market uses a mix of green and red baby leaf lettuce.

Response: Thank you for the comment. We agree that the desirable lighting treatment does not have to be the one that result in the highest yield. We discussed the effects of each light treatment on yield and quality, and growers can choose different treatments to strategically enhance nutritional quality or biomass.

I am also wondering if in conclusions authors can a sentence or two somewhat recommending that other red leaf lettuce cultivars should be screened with their recipes. It seems that there is a genotype x environment (light setting) interaction that could be considered by plant breeders while enhancing these compounds on red leaf lettuce.

Response: Thank you for your comment. We have added a recommendation in the conclusion ‘A broader range of lettuce cultivars should be evaluated under these EOP light treatments in future studies, as genotype × light environment interactions may influence the accumulation of anthocyanins and other beneficial phytochemicals’.

I would also like to request from authors to delete quotation marks from cultivar names when you precede the name with the word “cultivar”, “genotype”, etc. Congratulations on a well written manuscript.

Response: We have deleted the quotation marks from cultivar name that immediately followed “cultivar”, “genotype”. Thank you.

---

## [Decision Letter · Decision Letter 1]

From ultraviolet-B to red photons: Effects of end-of-production supplemental light on anthocyanins, phenolics, ascorbic acid, and biomass production in red leaf lettuce

PONE-D-24-54525R1

Dear Dr. Zhen,

We’re pleased to inform you that your manuscript has been judged scientifically suitable for publication and will be formally accepted for publication once it meets all outstanding technical requirements.

Kind regards,

Mayank Anand Gururani

Academic Editor

PLOS ONE

Additional Editor Comments (optional):

Reviewers' comments:

Reviewer's Responses to Questions

**Comments to the Author**

1. If the authors have adequately addressed your comments raised in a previous round of review and you feel that this manuscript is now acceptable for publication, you may indicate that here to bypass the “Comments to the Author” section, enter your conflict of interest statement in the “Confidential to Editor” section, and submit your "Accept" recommendation.

Reviewer #1: All comments have been addressed

Reviewer #2: All comments have been addressed

2. Is the manuscript technically sound, and do the data support the conclusions?

Reviewer #1: Yes

Reviewer #2: Yes

3. Has the statistical analysis been performed appropriately and rigorously? 

Reviewer #1: Yes

Reviewer #2: N/A

4. Have the authors made all data underlying the findings in their manuscript fully available?

Reviewer #1: Yes

Reviewer #2: Yes

5. Is the manuscript presented in an intelligible fashion and written in standard English?

Reviewer #1: Yes

Reviewer #2: Yes

6. Review Comments to the Author

Reviewer #1: (No Response)

Reviewer #2: Thank you for addressing my comments about your manuscript. I believe the manuscript is now ready for publication.

7. PLOS authors have the option to publish the peer review history of their article (what does this mean? ). If published, this will include your full peer review and any attached files.

**Do you want your identity to be public for this peer review?** For information about this choice, including consent withdrawal, please see our Privacy Policy .

Reviewer #1: **Yes: ** Nathan Kelly

Reviewer #2: No

---

## [Editor Report · Acceptance letter]

PONE-D-24-54525R1

PLOS ONE

Dear Dr. Zhen,

I'm pleased to inform you that your manuscript has been deemed suitable for publication in PLOS ONE. Congratulations! Your manuscript is now being handed over to our production team.

Kind regards,

on behalf of

Dr. Mayank Anand Gururani

Academic Editor

PLOS ONE